# Bend, Push, Stretch: Remarkable Structure and Mechanics of Single Intermediate Filaments and Meshworks

**DOI:** 10.3390/cells10081960

**Published:** 2021-08-02

**Authors:** K. Tanuj Sapra, Ohad Medalia

**Affiliations:** 1Department of Biosystems Science and Engineering, Eidgenössische Technische Hochschule (ETH) Zürich, 4058 Basel, Switzerland; 2Department of Biochemistry, University of Zurich, Winterthurerstrasse 190, 8057 Zurich, Switzerland

**Keywords:** lamin, vimentin, mechanobiology, cryo-electron tomography

## Abstract

The cytoskeleton of the eukaryotic cell provides a structural and functional scaffold enabling biochemical and cellular functions. While actin and microtubules form the main framework of the cell, intermediate filament networks provide unique mechanical properties that increase the resilience of both the cytoplasm and the nucleus, thereby maintaining cellular function while under mechanical pressure. Intermediate filaments (IFs) are imperative to a plethora of regulatory and signaling functions in mechanotransduction. Mutations in all types of IF proteins are known to affect the architectural integrity and function of cellular processes, leading to debilitating diseases. The basic building block of all IFs are elongated α-helical coiled-coils that assemble hierarchically into complex meshworks. A remarkable mechanical feature of IFs is the capability of coiled-coils to metamorphize into β-sheets under stress, making them one of the strongest and most resilient mechanical entities in nature. Here, we discuss structural and mechanical aspects of IFs with a focus on nuclear lamins and vimentin.

## 1. Introduction

In the crowded intracellular environment, information passes across distances up to tens of micrometers. One way to achieve this is through biochemical signals that rely on diffusion; another, more familiar way, akin to the macroscopic world, is physical transport [1]. Molecular motors walk on macromolecular tracks made of protein filaments, transporting vesicles, proteins and molecules from the cell membrane to the inside of the nucleus. Microtubules (~24 nm in diameter) and actin filaments (~8 nm in diameter) are examples of dynamic tracks that polymerize and depolymerize, enabling motor proteins to carry vesicular cargoes. Unlike microtubules and actin, intermediate filaments (IFs) are nonpolar structures, their assembly dynamics are nucleotide independent and they have no known transport function [2]. Cytoplasmic IFs assemble into ~10 nm diameter structures; however, lamin filaments are now shown to be ~3.5 nm [3].

IFs are the most versatile load-bearing entities found in nature that maintain the structural integrity of the cell and the nucleus [4]. Along with F-actin, IFs are key players in mechanotransduction [5]; mechanical wave propagation along the filaments is an ingenious engineering solution that cells have incorporated [1,6]. In simple words, the persistence length can be defined as the length of the polymer chain over which it stays rigid. As bending stiffness (*κ*_b_) and persistence length (*l*_p_) are directly related (*l*_p_ = *κ*_b/_*k*_B_T; *k*_B_ is the Boltzmann’s constant), out of the three cellular filaments, IFs are the most flexible with a *l*_p_ ~ 0.2–2 μm, while F-actin and microtubules are stiffer with a *l*_p_ ~10–20 μm and a *l*_p_ > 1 mm, respectively [3,7,8]. Persistence length is also a key parameter in determining the strain-stiffening of polymers [9], an important property in forming robust structures. Measurements of persistence lengths and load-bearing capacity in vitro show that microtubules and F-actin break under piconewton (pN) forces and do not absorb shocks. In contrast, IFs can be stretched to many times their original lengths and are able to resist failure even under nanoNewton (nN) forces [10,11,12,13].

The biochemical and immunological identities of IFs along with their functional importance have been well documented for over four decades [14]. IFs are encoded by at least 70 genes and are grouped into six types (I–V, and orphan); the most commonly studied are keratins (types I and II), vimentin, desmin (type III), neurofilaments, nestin (type IV), lamins (type V), phakinin and filensin (orphan) [15]. Lamins are the only IFs that form the nuclear lamina, a shell-like meshwork abutting the inner nuclear membrane, and impart a structural and mechanical framework to the nucleus [16,17]. The cytoskeleton is connected to the nucleoskeleton–lamin meshwork-via LINC (linker of nucleoskeleton and cytoskeleton) complex proteins that traverse the double bilayer membrane of the nucleus creating a continuous network [18]. This molecular continuum is a functional necessity to convert mechanical signal input from the extracellular environment into genetic outputs inside the nucleus [1,19,20]. Thus, cytoskeletal and nucleoskeletal networks are physically fine-tuned to achieve the required cellular functions [21,22]. Mutations in any one filament type or associated proteins may perturb the delicate balance leading to debilitating diseases [8].

More than 110 distinct diseases are associated with IFs, affecting almost every organ in the human body (http://www.interfil.org/, accessed on 1 August 2021) [23,24]. The diseases are caused by mutations in multiple cell types—lamins in nucleated cells, keratin in epithelial cells, vimentin in mesenchymal cells, neurofilaments in neurons, desmin in myocytes [25] and filensin and phakinin in lens [26]. Keratins and vimentin are also used as clinical markers for cancer [27,28,29,30], while vimentin is suggested to participate in SARS-CoV infection [31]. Lamins are known to be key to cancer progression, ageing and laminopathies [32,33]. Over 500 reported mutations in the lamin genes—some recently found in *LMNB* [34] and many studied in *LMNA*—are implicated in multiple overlapping clinical phenotypes of four major disease types: striated muscle diseases, lipodystrophy syndromes, peripheral neuropathy and accelerated ageing disorders [35,36]. This group of diseases, termed laminopathies, is an important example of mutations affecting load-bearing tissues, such as striated muscles, potentially leading to mechanical failure. Due to their importance in cell physiology [37,38], great efforts have been focused on understanding the mechanical role of IFs.

## 2. IFs Structure, Lamin Filaments and Network Organization

Besides metazoan cells, IFs have been investigated structurally and mechanically in a number of invertebrate model systems including *Caenorhabditis elegans (C. elegans)* [39], *Danio rerio* [40], *Xenopus laevis (X. leavis)* [41] and, *Drosophila melanogaster* [42], enhancing our functional understanding of this important class of cytoskeletal proteins. In general, insights into the structure of IFs have been few and far between [43,44,45]. A few examples of successful attempts include vimentin and keratin filaments by cryo-EM [46,47,48] and X-ray structures of the 2B subdomain [49] and the α-helical coiled-coil region of vimentin [50], the 2B subdomain of lamin A [51] and heteropolymeric keratins K5 and K14 [52]. Additional biophysical approaches were also utilized to resolve the structure of IF fragments, summarized in [53].

All IFs share a common tripartite domain structure comprising a highly conserved, centrally located α-helical coiled-coil rod domain, capped by a flexible N-terminal head domain and a flexible C-terminal tail domain. Lamin filaments are type V IFs, and therefore share the conserved tripartite structure with other IFs. In aqueous solution, lamins are constitutive dimers of α-helices arranged to form a coiled-coil rod domain with an N-terminal head domain (31 amino acids, aa) and a C-terminal tail (196 aa) that contains an Ig domain. Thus, the formed polymers are decorated with globular Ig domains along the filament with ~20 nm spacing [3]. In mammals, the lamin meshwork mostly comprises four types of lamin isoforms—lamins A and C, and lamins B1 and B2 [54]. The first snapshots of the lamin meshwork were obtained in the early 1980s using platinum-shadowing electron microscopy showing the LIII filaments that constitute the majority of lamins in the *X. laevis* oocyte nucleus [41]. The images of fixed samples showed the lamin filaments arranged in an almost perfect orthogonal pattern without nuclear pore complexes (NPCs). Using field-emission scanning electron microscopy, a similar arrangement was observed later [55]. The diameter of a single lamin filament was refined to be ~7 nm (estimated to be 3–4 nm without the chromium metal coat).

Obtaining structures with atomic details has been a daunting challenge for IFs in general, owing to their insoluble nature and a tendency to form paracrystalline assemblies in vitro [56]. A ‘divide and conquer’ strategy has employed truncated filament fragments for X-ray crystallography, electron paramagnetic resonance and electron microscopy, allowing the creation of atomic models of full-length filaments [45,57]. For the most part, IF models have proven useful; however, the accuracy of structures obtained by stitching together constituent parts, that may behave very differently in crystals without their native molecular contacts, has pushed for efforts to explore better techniques for obtaining native structures in situ or even in vivo. Recently, 3D-structured illumination microscopy (3D-SIM) in conjunction with image processing analysis provided a view of the lamin meshwork inside the mammalian nucleus [58,59].

Unprecedented advances in cryo-electron microscopy have unleashed an era of structural determination of macromolecules at atomic resolution [60,61]. Besides the resolution revolution in single particle imaging, in situ structural analysis by cryo-electron tomography (cryo-ET) provides the means to gain structural insights into intact cells [62] (Figure 1). Cryo-ET has been used to reveal the structural make-up of whole organisms, cells and cellular organelles, and cellular processes [63,64,65,66]. Initial cryo-ET images of ectopically expressed *C. elegans* lamin meshworks in *X. laevis* oocyte nuclei showed that 4–6 nm thick filaments formed varied geometries unlike the orthogonal patterns observed before [67]. In vitro, *C. elegans* lamins assemble into 10 nm thick filaments and form paracrystalline arrays [68]. However, the relevance of such assemblies in vivo is unclear. Nevertheless, paracrystals do offer an opportunity to study mutations. For example, in *C. elegans*, the Hutchinson-Gilford Progeria syndrome-causing mutation Q159K in coil 1B (E145K in lamin A) led to a different assembly of only two protofilaments instead of three or four as in the wild-type lamin [69]. Similarly, the mutation L535P in the tail domain (L530P in lamin A), associated with Emery-Dreifuss muscular dystrophy, led to abnormal paracrystal assembly. Cryo-ET snapshots have afforded a high-resolution comparison in wild-type and mutated paracrystalline assemblies [70].

In recent years, technical developments in cryo-focused ion beam (FIB) milling of cells in conjunction with cryo-ET have provided comprehensive insights into the structure and organization of protein complexes in vivo and in situ [63,64]. Insights into the molecular arrangement of lamins and NPCs in an intact mammalian nucleus were recently obtained by using ghost nuclei devoid of much of the chromatin [3] (Figure 1A). The approach allowed observation of the structural framework of the mammalian lamin meshwork and measurement of the basic physical dimensions of single lamin filaments. Surprisingly, the ancestor of all ~10 nm thick IFs assemble into ~3.5 nm thick filaments [71].

Recent studies suggest that anti-parallel coiled-coil dimers are the building blocks for further polymerization. The study used a bacterially expressed human lamin A/C fragment (~38 nm long, full length is ~54 nm) where high-resolution X-ray structure and cross-linking mass spectrometry analysis were utilized [72,73]. Thus, current knowledge suggests that lamin assembly is initiated by polypeptide dimerization through α-helical coiled-coils to form staggered head-to-tail polar filaments [74,75]. The staggered filaments interact laterally to form tetrameric protofilaments, and finally interconnect to form the lamin meshwork [3,76].

Labeling lamins with colloidal gold nanoparticles (6 nm and 10 nm) showed that lamin A/C and lamin B1 filaments exist in segregated domains [3]. Analysis of fluorescence images obtained by stochastic optical reconstruction microscopy (STORM) has suggested that lamin A and lamin B meshworks are concentric; lamin B is anchored to the inner nuclear membrane via a farnesyl moiety and the lamin A meshwork is perched on it [77]. It remains to be verified if the differences in localization are independent of sample preparation which included detergent and chemical fixation procedures. Nevertheless, arrangement of the two segregated meshworks may have far reaching implications in diseases where blebbing is a commonly observed phenotype [78,79]. Recent cryo-ET findings suggest that mutations in *LMNA* (Lmna^H222P/H222P^) in mouse embryonic fibroblasts perturb the meshwork organization [76] (Figure 1B). The mutation has been shown to cause mechanical aberrations, i.e., increased stiffness, of muscle cells [80].

## 3. In Situ Mechanical Probing of IF Networks in Cells and Nuclei

Owing to their mechanical roles in the cytoplasm and the nucleus, IFs have been the focus of intense mechanical investigations [81,82]. As in structural studies, *X. laevis* oocyte nuclei proved instrumental for mechanical studies of nuclear lamins. Ectopically expressed lamin A in *X. laevis* oocyte nuclei forms a distinct thick layer (30–100 nm) on top of the endogenous lamin LIII meshwork [83]. Mechanical studies with the atomic force microscope (AFM) measured the stiffness values of oocyte nuclei that increased from ~1–2 mN/m (control without lamin A) to ~5–7 mN/m (ectopically expressed lamin A). The stiffness increase was dependent on the expression level of lamin A as the nucleoplasm of *X. laevis* oocyte nucleus is suggested to be soft. Measurements on isolated nuclei using micropipette aspiration converged to a value of 25 mN/m for the elastic modulus of the lamin network [16]. The E145K mutation in lamin A, known to cause Hutchinson-Gilford progeria syndrome (HGPS) [84], leads to abnormal clustering of centrosomes, mis-localization of telomeres and perturbs the lamin network as indicated by lobulated nuclei [70]. Similar to wild-type lamin A, the E145K mutation containing lamin A expressed in *X. laevis* oocyte nuclei assembled as 100 nm thick multi-layered sheets on top of the endogenous lamin LIII meshwork [85]. AFM stiffness measurements of the E145K-lamin A-expressing nuclei indicated values of ~5–9 mN/m, similar to wild-type lamin A, but higher than lamin LIII. A more striking difference was observed in the Young’s moduli of lamin A (~40–80 MPa) and E145K-lamin A (~100–340 MPa) [85].

Direct evidence for lamin alteration by the E145K-lamin A mutation came from mechanical measurements of isolated nuclei of human dermal fibroblasts [86]. A comparison of the nuclei of a 4-year-old progeria patient (E145K-lamin A) and two healthy donors of 10 and 61 years of age showed a clear increase in stiffness for progeria-expressing nuclei (800 μN/m) and the aged donor nuclei (600 μN/m), as compared to the young donor nuclei (300 μN/m). Interestingly, intact fibroblasts from the three donors showed no clear difference in nuclear stiffness, presumably due to the sensitivity limitations imposed by the cantilever that could not detect differences in nuclei surrounded by a buffering cushion of the cytoplasm, the cytoskeletal network and the cell membrane.

Recently, an in situ correlative approach was devised to study nuclear mechanics using AFM and confocal microscopy. A sharpened AFM cantilever tip was inserted into the cell nucleus through the plasma and the nuclear membranes at forces > 5 nN [87]. Force-extension (FE) curves (a plot of deformation/extension of an object under force) and confocal microscopy images enabled a correlation of the breaching of the cell and nuclear membranes and pushing of the lamina. FE profiles clearly showed points when the cell membrane and the nuclear membrane were breached. The Young’s modulus of isolated nuclei (~9 kPa) was three-fold lower than that of nuclei within the cell (~27 kPa). The difference may owe its origin to the exposure of the nucleus to intracellular balancing forces in vivo, a cornerstone of the tensegrity model [88]. Such new methods may also find applications in medical diagnostics. For example, comparing the Young’s moduli of two bladder cancer cell lines showed that the isolated nuclei from low metastatic cells were stiffer (~8 kPa) than those from the high metastatic cells (~6 kPa). The same trend was observed for intact cells. A possible explanation forwarded was the difference in the lamina composition and lamin A concentration that imparts stiffness to the nucleus [87]. Studies utilizing spherical AFM tips gave a Young’s modulus of 0.4 kPa in mouse embryonic fibroblasts (when measured on top of the nucleus) [89]. The discrepancy in values is not surprising as force measurements depend on numerous factors, such as the cell type, passage number, cantilever tip geometry, contact area between the tip and the sample surface and the speed of the tip, to name a few [90]. As the contributions of other cellular components are difficult to separate in such studies, minimal systems are required to rule out effects arising from the nucleoplasm, nuclear membrane and chromatin. In vitro measurements of assembled filaments, or direct measurements of the meshwork and filaments in situ are required to unravel the mechanics of IFs in the absence of other cellular influences.

Other IFs, such as keratins, are important mechanical building blocks of numerous biological structures in a variety of cell types and tissues [91]. The keratinocyte is an important cellular model system for studying the role of keratins in epithelial tissues. Pushing an AFM cantilever on keratinocytes and manipulating superparamagnetic beads inside a cell by magnetic tweezers showed that deleting type I or type II keratins leads to cell softening. The Young’s modulus decreased to 412 Pa from 752 Pa in wild-type cells, and the cytoplasmic viscosity was reduced by > 40% in keratin-deficient cells. [92]. Stiffness recovery was achieved upon re-expressing the K14 and K5 keratins in type I and type II keratin-deficient cells, respectively. Similarly, a 60% increase in cell deformability was observed by non-contact measurements in a microfluidic optical stretcher with a dual-beam laser trap [93]. Interestingly, the K14 keratin cytoskeleton is deformed less when keratinocytes are on stiff substrates (70 and 214 kPa) compared to a soft one (8 kPa). The Young’s modulus (stiffness) showed a progressive increase with substrate stiffness [94]. Similarly, Young’s moduli of cells with and without vimentin also depend on substrate stiffness [95]. Substrate-dependent changes in the stiffness of nucleus correlated to up- and down-regulation of lamin A and lamin B have also been observed [96].

## 4. In Vitro Single-Molecule Mechanical Characterization of IFs

IF mechanical properties have been characterized using various techniques which measure the ‘bulk’ properties of network gels [7]. Micro-rheology, in particular, has given a foundation to understand the mechanical response of IF networks [97,98], and the physical origin of strain-stiffening under shear stress [9,99]. Under low stress, the polymers exhibit reversible linear entropic stretching. As the polymers are stretched, the chains undergo irreversible enthalpic unfolding resulting in nonlinear strain stiffening [100]. Our understanding of the mechanical properties of IFs and their physical origin has reached a consensus; however, the specific properties of different IFs remain an active area of investigation.

The first experimental proof that IFs are flexible and extensible came from stretching hagfish slime threads comprising keratin-like filaments using a home-built device [101]. The bundles were found to retain elasticity up to strains of 34%, were capable of extending 2.2-fold, and exhibited a high tensile strength of 180 MPa and toughness of 130 MJ/m^3^. In the same study, α-helix to β-sheet transition of the filaments was reported based on X-ray diffraction and congo-red staining experiments. Single-molecule stretching of IFs was tested by AFM measurements of recombinant murine desmin [11,102], recombinant keratin and rat brain neurofilaments [11] and vimentin [103,104]. In these early studies, filaments adsorbed onto a surface were dragged by an AFM cantilever applying forces of 30–40 nN resulting in stretching of IFs up to 4.5-fold before breaking.

Desmin, a type III IF, physically connects successive sarcomeres and adjacent myofibrils in muscle cells. Their structural role exposes desmin filaments to repetitive extension and contraction, making them attractive targets for mechanical characterization. In vitro, desmin filaments are able to withstand forces up to 3.5 nN, possess a tensile strength of 240 MPa, dissipate mechanical energy (50 MJ/m^3^) and achieve strain hardening above 50% extension, reaching up to 240% [102]. Vimentin, another type III IF, assembles into a dense network in cells of mesenchymal origin. This network spans much of the cytoplasm, from the cell membrane to the outer nuclear membrane where they connect to the LINC complex, and therefore to the lamin meshwork [105]. The vimentin filament network forms a protective cage around the nucleus, conferring mechanical stability during migration through micrometer-sized pores [89]. The nanometer tip of the AFM cantilever was used to push single recombinantly expressed vimentin filaments freely suspended over a porous alumina membrane and record the filament extension. A Young’s modulus of 300–400 MPa was determined presuming axial sliding between subunits. The modulus increased to 900 MPa when the filaments were stabilized with glutaraldehyde, preventing sliding but permitting bending [103]. These results are in agreement with molecular dynamics (MD) simulations of stretching vimentin dimers that estimate a Young’s modulus value of 380–540 MPa [106].

## 5. Behavior of Single Vimentin Filaments at Controlled Force

Albeit surface effects are difficult to rule out, in vitro AFM studies have provided relevant mechanistic insights into IF mechanical properties [104]. Like AFM, laser optical trapping (LOT) provides a sensitive approach to measure the forces required to stretch filaments that are anchored between micrometer-sized beads, held by lasers acting as soft springs [107,108] (Figure 2A). A force of ~600 pN was measured to stretch single, in vitro assembled vimentin filaments up to a strain of ~125% along the long axis (Figure 2B,C). Similar to MD simulations [106], three clear regimes were observed in the FE profile: (i) an initial linear increase reaching a force of ~300 pN denoting the elastic response of the α-helical coiled-coil at a strain of ~10%; (ii) a plateau where the α-helical coiled-coils unfold and begin to transition to β-sheets; (iii) another linear rise that signifies stiffening at strains >100%. Interestingly, at low speeds, single vimentin filaments behaved as a soft material capable of extending up to a strain of 125%. As the stretching speed was increased, the stiffening occurred faster and the filaments were extended only 75% [108]. This behavior may reflect different states of the vimentin dimers comprising the tetrameric filament used in the experiments. Nevertheless, a two-state model was invoked to capture the α-helix to β-sheet transition of the dimers during filament stretching (Figure 2D). 

A single vimentin filament is a shock absorber capable of dissipating 70–80% of the input energy [109]. However, this property changes as single filaments become softer with every stretching cycle. The physical origin of this behavior may arise from the unfolding of a few α-helical coiled-coils while others may still be folded. As more coiled-coils are unfolded with repeated stretching, the filaments may progressively soften. This model also captures the observation that the α-helical coiled-coil domains of vimentin filaments do not unfold reversibly when stretched at low forces even if allowed to rest for up to an hour [109,110]. Interestingly, chemical cross-linking of the filaments with paraformaldehyde and glutaraldehyde made the α-helix unfolding reversible even when no time is allowed between stretching–relaxation cycles. The potential barrier also widened from 0.17 nm for native vimentin to 1.2 nm for glutaraldehyde-treated vimentin, indicating that the molecular mechanism of α-helix unfolding leading to β-sheet formation is different in the two cases. One proposed mechanism is, that in a chemically cross-linked filament, parallel helices unfold together, whereas in an untreated filament, the stochastic unfolding of an α-helix destabilizes the parallel helices [109]. Cross-linking the polypeptides in a filament bypasses the intermediate state and the filaments are capable of recuperating to their original stiffness. The two-state model was aptly revised to a three-state model by introducing an intermediate state en route to the β-sheet from the α-helix state (Figure 2E,F). Based on these results, it has been suggested that once unfolded, vimentin is incapable of functioning properly and may require repair mechanisms such as subunit exchange or reassembly processes [110]. Whether the filaments respond to forces in the same way when part of the intricate intracellular cytoskeletal network remains to be investigated. Surprisingly, the mechanical properties of single lamin filaments were shown to depend on the lamin meshwork topology or connectivity (discussed below) [111].

To capture the full mechanical response of vimentin filaments exceeding forces of 800 pN, AFM was used to pull putative single filaments from a mica surface using a cantilever tip [108]. At low forces (~600 pN), FE profiles similar to optical tweezers were measured. Beyond this, the plateau reached up to 200% extension and a steep linear rise in force up to an extension of 300% was measured. The load-bearing capacity of vimentin filaments without breaking was measured to be at least 8 nN. Similar to vimentin, when single keratin filaments are stretched between 0.3 μm/s and 2.5 μm/s, LOT shows an FE profile with an initial elastic regime at low strains, followed by a plateau for strains between 20% and 80%, and a further steep rise up to 800 pN leading to filament stiffening [112].

The structural differences between the major cytoskeletal filaments allow a cell to respond to force signals. Ion concentration, pH and post-translational modifications provide another mechanism to fine-tune the mechanical response of the filaments. For example, the mechanical responses of both keratin and vimentin filaments change drastically at a high ion concentration (100 mM KCl); keratin responds monotonically to an increasing strain whereas vimentin shows a nonlinear response with three clear regimes. Moreover, ~50% higher energy is required to stretch vimentin compared to keratin in high ion medium. The change in response has been explained by a difference in the negative charge per unit length of the two proteins. Vimentin carrying a higher charge (19 *e*/monomer) compared to keratin (8.5 *e*/monomer) requires more energy for stretching due to increased charge screening at high ion concentration. Concurrently, in vitro rheological measurements on vimentin networks showed that divalent ions, such as Ca^2+^ and Mg^2+^, tend to cross-link the filaments and increase their stiffness [113,114]. Truncation experiments suggested that the cross-linking is most likely mediated by the last 11 amino acids of the carboxy-terminal tail domain. Interestingly, the tail domain carries four negatively charged residues and two histidines. Ion concentration and also pH may fine-tune the delicate interplay between the complex IF networks in the cell in response to the barrage of forces [115].

Phosphorylation of IFs has a key role in cell division, migration, adhesion, cell differentiation and cell death [116]. The mechanical response to the stretching of vimentin filaments partially phosphorylated by protein kinase A was recently investigated in an optical trap [117]. Increasing phosphorylation (1%, 5%, 10%) made the filaments softer as observed in the FE profile and denoted by the Young’s moduli. The softening is owed to the negative charges introduced by phosphorylation that weaken the inter-dimer coupling mediated by the positive charges on the head domain. Binding of the regulatory protein 14-3-3 to the phosphorylated filament further softened the filament. Although the exact implications are not clear yet, this result may suggest a role in assembly–disassembly of the filaments, cell migration and metastasis [118,119]. Phosphorylation effects on the mechanical properties of lamin [120] and keratin [121] require further investigations.

## 6. Interrogating Single Lamin Filaments In Vitro, In Silico and In Situ

Membranes of nuclei hosting mutant lamins or devoid of lamins are prone to frequent ruptures, resulting in DNA damage during cell migration through confined spaces (<20 μm) [122,123,124,125]. Nuclear mechanics and its most important determinant—lamins—have therefore been studied for many years. Techniques such as micropipette aspiration and stretchable substrates give information on deformation characteristics of the lamin meshwork in the whole nucleus [126,127] and in the context of the whole cell [128], respectively. Compared to other IFs, in vitro mechanical characterization of single lamin filaments is lagging due to difficulties in expressing native-like filaments and their tendency to form paracrystalline assemblies. Lamin paracrystals most likely represent kinetic intermediates on the energy landscape and not native functional forms. The mechanical properties of filaments in paracrystals could be wildly different from those in vivo, where lamin dimers form thin filaments and a mesh-like network [54,129]. Nevertheless, paracrystalline assemblies provide an alternative approach to decipher the biophysical properties of lamins [12,130].

The majority of the mutations specific to lamin A cause debilitating diseases [131]. While there has been no success in obtaining the complete lamin A filament in the soluble form, its rod domains, 1B and 2B, have been expressed individually. AFM single-molecule force spectroscopy (SMFS) of these mini-lamin-forming domains provided insight into their roles in the elasticity of the lamin meshwork [132]. The average unfolding forces of domains 1B and 2B were ~70 pN and ~80 pN, respectively. Rod domain 1B was found to be a key player in meshwork formation and the major load-bearer, resisting strains up to 200% compared to 100% by domain 2B. A network of full-length lamin filaments is capable of withstanding strains up to 500%, suggesting that the response to force is an emergent property of the lamin meshwork [111]. In vitro studies as described above for vimentin are required on full-length or components of lamins for a holistic view inside the nucleus. As shown for vimentin filaments that soften upon binding of 14-3-3, the effects of Lap 2α, Lap 2β and lamin B binding protein on lamin mechanical properties are also worth investigating.

As in studies of vimentin, in silico efforts have bridged the gap in knowledge between the mechanical behavior of single lamin dimers [133] and in a meshwork [134]. Molecular snapshots of simulated stretching showed that the α-helical coiled-coils unfold at low forces (a few hundred pN), before a structural transition of α-helix to β-sheet, leading to stiffening at a high force (a few nN). Finally, β-sheets undergo a stick-slip mechanism leading to a breakdown of the filament at ~6 nN. Cell migration, division and diseases expose the lamin meshwork to continuous stretching and pushing forces in vivo [135,136]. Network models based on cryo-ET data were created and the pushing of lamin filaments in the models was simulated to garner molecular insights. Corroborating the in-plane stretching of lamin filaments [134], pushing lamin filaments orthogonal to the meshwork showed unfolding of the α-helical coiled-coil domain at low forces, α-helix to β-sheet transition and stiffening at high forces, and finally filament failure [111].

Direct measurements of lamins—in vitro or in situ—have been lagging owing to technical challenges; for example, expressing and assembling native filaments in vitro and measuring single filaments in situ using sensitive tools [87,137,138]. Recently, an integrative in situ approach was employed by combining AFM, cryo-ET and MD simulations to understand the design principles of the lamin meshwork in the nuclei of two metazoan classes—Amphibia and Mammalia [111]. Owing to its large size (~400 μm) compared to the mammalian nucleus (~20 μm), the *X. laevis* oocyte nucleus was chosen to mechanically characterize lamin filaments. This was the first time lamin filaments were characterized in close-to-native conditions, i.e., in its meshwork attached to the inner nuclear membrane in physiological buffer conditions (Figure 3A). The tip of the AFM cantilever was positioned at random positions on lamin filaments and pushed at a constant velocity to determine the force tolerance of single filaments. The similarities in the experimental and simulated FE profiles and the failure forces suggested that the molecular mechanisms were most likely the same (Figure 3B,C). As in the simulated FE profile, lamin filaments exhibited a nonlinear stretching behavior: at a low pushing force (≤0.5 nN), the filaments showed reversible unfolding of α-helical coiled-coils, and as the force increased, stiffening of filaments attributed to α-helix to β-sheet transition was observed (Figure 3D). For comparison, reversible stretching of recombinant vimentin filaments has been observed up to a pushing force of 0.13 nN [103] and more recently for chemically cross-linked vimentin [110].

When subjected to continuously increasing forces up to 5 nN, lamin filaments deformed to ~90 nm before failing in steps of 4 nm and 8 nm. Force-clamp experiments, where single filaments in a meshwork were subjected to a constant force (0.75–3 nN), showed that the filaments mainly failed in discrete steps of 1, 4 and 8 nm [111]. A continuous increase in the stiffness values was observed; the α-helical coiled-coils are soft with a stiffness of 0.008 N/m and β-sheets are stiff with values > 0.3 nN/nm. Desmin filaments have a comparable initial stiffness of 0.006–1.2 N/m [102]. The broad distributions in force and stiffness observed for lamins in situ could be due to the varied mechanical strength of the filament crosslinks in the anisotropic meshwork [139]. It is possible that the failure force of a lamin filament depends on its location in the meshwork and the position where it is loaded, i.e., close to a connecting node or away from it. The physical origin of this phenomenon is also reflected in a purely homogenous meshwork, such as a defect-free graphene lattice where the force distribution is very narrow [140]. Similarly, forces required to break single threads in a spider’s web also depend on the type of the thread and the structure of the web [141].

Absorbing shocks is proposed to be a key function of IFs [16,109]. The α-helical coiled-coil domain of lamin filaments unfolds reversibly up to a strain of 120% absorbing ~10^−17^ J. Pulling collagen molecules adsorbed on a glass surface or directly from the bone femur resulted in the breaking of ‘sacrificial’ bonds in the sub-nanoN range. The energy of dissipation of such sacrificial bonds was measured to be ~10^−17^ J [142]. Similar to collagen, lamin and other IFs may have evolved protective mechanisms whereby α-helix unfolding absorbs energies in the mechanical range (up to 500 pN) that most physiological processes operate. The energy absorbed increases to ~10^−16^ J during the entire unfolding, stiffening and failure of a lamin filament [111]. This is noteworthy because the energy is equivalent to breaking ~170 C–C bonds, but the filament failure force of ~3 nN corresponds to the breaking of a single covalent bond [143]. It can therefore be surmised that no covalent bonds are broken during the process but the energy disrupts non-covalent interactions (sum of charges, van der Waals forces, and hydrogen bond interactions) during the unfolding of the filament domains, α-helix to β-sheet transformation, and is stored in the β-sheets [109].

## 7. The Role of Network Topology in Lamin Mechanics

Network characteristics and mesh-size are recurring phenomena in a cell and changes in those are markers of disease [144]. For example, in situ particle-tracking micro-rheology of keratin networks suggested the influence of mesh size and possibly connectivity on the mechanical properties of the network under shear stress [145]. Cryo-ET data reconstructs the 3D volume of a network from images captured in different planes [146]. The networks can be analyzed using graph theory principles to obtain quantitative parameters of the network topology, i.e., a connectivity map akin to a subway chart [147]. Simulating lamin filament pushing showed that increasing the connectivity of the network increased the strength, extensibility and toughness of single filaments [111]. This may be a straight-forward yet intriguing result that can be invoked to explain various observations related to changes in nuclear morphologies and associated dysfunctions in lamin-associated diseases, as well as possibly other IF-associated disease. For instance, the lamin meshwork forms ordered domains in HGPS patient cells, distorting the nuclear morphology and making it ‘brittle’ [148]. This concurs with the structural hypothesis that mutations cause aberrant changes in the mechanics of the lamin filaments with deleterious effects on the lamin meshwork and the nuclear envelope [149]. However, this does not rule out the gene-regulation hypothesis that mutations in the lamin protein affect crucial protein binding sites, thereby leading to nuclear abnormalities. It is likely that both structure and gene-regulation are affected and are not exclusive [150]. Breaks in connections between lamin filaments and SUN-KASH proteins, cytoskeletal filaments and nuclear membrane proteins may also perturb the lamin network topology [151].

## 8. The Road Ahead

The structural framework of cells has evolved to respond efficiently to different mechanical shocks and traumas. IFs are excellent examples of this phenomenon—even different isoforms of the same filament may have different roles. The first line of defense is the actin cortex abutting the cell membrane, providing a stiff, although brittle, layer under high loads. Once parts of the actin cortex are breached, the force reaches the underlying layers of IFs that are stiff and absorb shocks at low forces. However, at higher loads, IFs start to stiffen before succumbing. In the case of lamins, load-tolerance was shown to increase with stiffness [111]. A few mechanical properties of IFs are summarized in Table 1.

A low force tolerance material such as F-actin forms a cortex around the cell, acting as a sensitive sensor for miniscule pN forces relayed from the cellular environment via the integrin receptors. The structure of IFs bestows superb mechanical properties on them—long rope-like filaments that form networks, α-helical coiled-coils capable of sliding when stretched, strain-induced stiffening at high forces. It is intriguing, though perhaps not surprising, that despite the differences in the dimensions and sequence identities of vimentin and lamin, their mechanical characteristics are comparable [152]. It would be worth investigating the entire mechanical regime of vimentin up to nN forces in situ or in a close-to-native environment. Possibly, similar to lamin filaments, vimentin filaments will portray a network-dependent enhancement of their mechanical properties [114].

The concentrations of different filaments are fine-tuned depending on the cell-type. For example, lamin concentrations are regulated depending on the development stage and cell type; arteries and eye lens have cross-linked vimentin. Lamin A is predominantly present in ‘harder’ fibroblasts and muscle cells, and the lamina thickness depends on the cell type and the physiological state of the cell [153]. The amount of lamin A has been shown to scale with tissue stiffness [96]. Similar observations were recorded for keratin networks that undergo structural remodeling on stiffer substrates. Depending on the substrate stiffness, keratin may play an important role in modulating A-type lamin expression and mechanotransduction to the lamina [94]. It may imply that lamin expression levels in different tissues dictate the topologies of the nuclear lamin, further influencing their mechanical properties and the ability to protect the nucleus from extracellular forces.

The effect of different mutations (in the rod and tail domains) on the lamin meshwork architecture is an ongoing area of investigation. A recent study suggests that the H222P mutation in lamin A of mouse embryonic fibroblasts does not lead to a marked change in the lamin meshwork, although nuclear volume and chromatin organization are affected [76]. Detailed investigation is required to understand at what stage of assembly, i.e., dimer, tetramer or higher-order assemblies, do mutations cause mechanical changes.

In vitro experiments provide a good starting point to study individual elements. However, these experiments do not capture an accurate picture of IF properties in many cases, as demonstrated by paracrystalline lamin assemblies. The interplay between the different filaments and their fine tuning of cellular signaling and mechanical protection is far from clear. Ideas from bottom-up synthetic biology can be applied, for example, to mimic the nucleus and increase system complexity by reconstituting essential processes [154]. These are challenging ventures and require new approaches and techniques to completely understand the functional make-up of these complex systems. We envision that combining imaging and nano-manipulation methods and tools (e.g., AFM, LOT, cryo-EM, super-resolution microscopy, MD simulations) will open avenues to correlate the structure and mechanics of cyto- and nucleoskeletal networks in situ.

## Figures and Tables

**Figure 1 cells-10-01960-f001:**
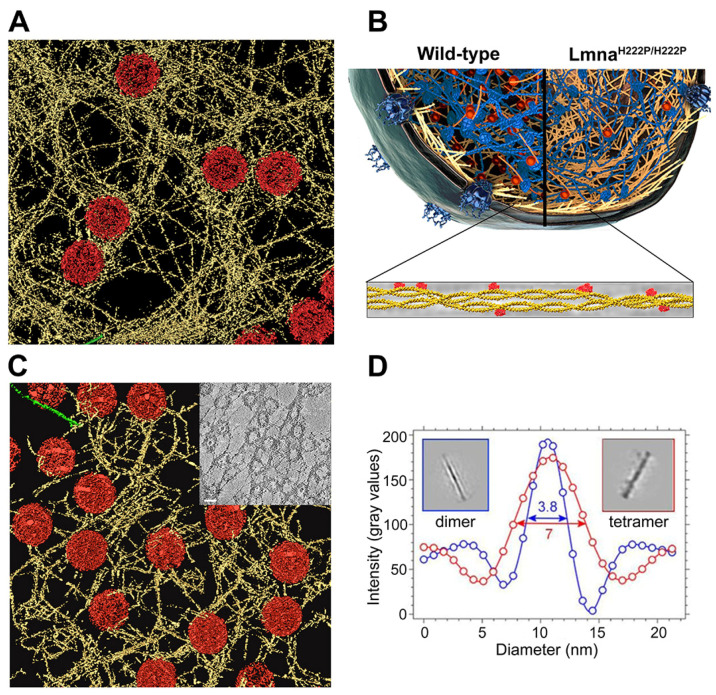
Molecular assembly of nuclear lamina and structure of a lamin filament. (**A**) Surface rendered view of a cryo-electron tomogram of mouse embryonic fibroblast nucleus. Nuclear lamins formed a 3D meshwork of filaments (yellow) connected to the nuclear pore complexes (NPCs, red). Actin filament is shown in green. Field of view, 700 nm × 700 nm [1]. (**B**) A model showing the wild-type nuclear lamina (lamin filaments in yellow) connected to chromatin (blue) and chromatin binding factors (red), and the changes in mutant (Lmna^H222P/H222P^) mouse embryonic fibroblasts where lamin meshwork is more exposed. A lamin filament tetramer comprised of two α-helical coiled-coils with Ig domains (red); figure adapted from [2]. (**C**) Surface rendered view of a cryo-electron tomogram acquired on a spread nuclear envelope of the *X. laevis* oocyte. Inset, a 10 nm slice through a tomogram of spread NE showing the nuclear lamina of *X. laevis* oocyte; scale bar 100 nm [3]. (**D**) Density of the rod-like structures from cryo-electron tomograms of two structural class-averages (framed blue and red) indicating filament thickness of ~3.8 nm and ~7 nm in agreement with a tetramer and lateral association of tetramers, respectively; figure adapted from [3].

**Figure 2 cells-10-01960-f002:**
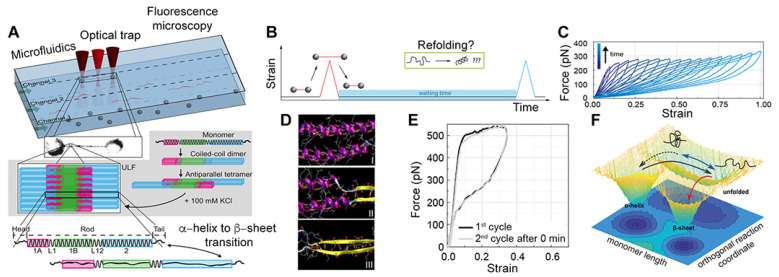
Stretching vimentin filaments in vitro. (**A**) Top, set-up combining microfluidics and laser optical traps for stretching single vimentin filaments. Single vimentin filaments flowing through microfluidic channels were captured and attached to micrometer-sized beads held in optical traps (shown in grey scale zoom). Middle, the assembly of vimentin is shown starting from a monomer to the unit length filament. Bottom, α-helical coiled-coil transitions to a β-sheet under applied strain; figure adapted from [4]. (**B**,**C**) Experiment protocol showing stretching, waiting and refolding of single vimentin filaments. Repeated stretching and relaxation of a vimentin filament to increasing lengths showed progressive softening without refolding of the structure. (**D**) Molecular dynamic simulation snapshots of α-helical to β-sheet transformation during vimentin stretching. I, II and III denote the α-helical coiled-coil, unfolding and conversion to β-sheets, respectively; figure adapted from [5]. (**E**) Chemical cross-linking of vimentin by glutaraldehyde permitted filament refolding apparently to its initial state [6]. **(F**) A three-state model captures the unfolding and α-helix to β-sheet transition of vimentin through an intermediate state. Refolding is only possible from the α-helical unfolded state. Once the β-sheet is in the energy well, there is no going back to the α-helical state. (**B**,**C**,**E**,**F**) reprinted with permission from [6]. Copyright 2019 American Chemical Society.

**Figure 3 cells-10-01960-f003:**
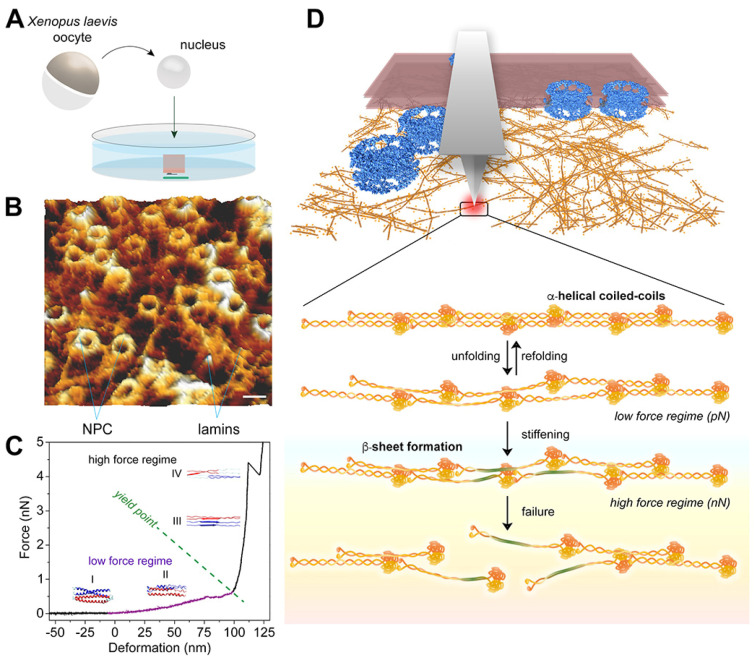
Mechanical characterization of lamin filaments in native meshwork. (**A**) Schematic illustration of the experimental set-up. Nuclei isolated from *X. laevis* oocytes were attached onto a poly-L-lysine-coated glass dish, and dissected to remove the chromatin and image the nuclear lamina [3]. (**B**) High-resolution AFM image of the nuclear lamina from the nucleoplasmic side showing areas of orthogonally arranged lamin filaments interspersed with NPCs; scale bar 100 nm. (**C**) A typical force-extension (FE) signal showing the nonlinear behavior of a lamin filament in the meshwork. A single lamin filament in the meshwork when subjected to mechanical push showed a low force regime and a steep rise leading to failure of the lamin filament. The different regions were assigned to the molecular changes in the lamin α-helical coiled- coils: I and II denote coiled-coil unfolding, III denotes α-helix to β-sheet transition, and IV denotes the failure of β-sheets. The yield point denotes the point of plastic or permanent deformation, i.e., irreversible structural change. (**D**) Schematic model showing the response of lamin filaments when subjected to different levels of external force in vivo [1,3]. Figures adapted from [2,3].

**Table 1 cells-10-01960-t001:** Mechanical properties of intermediate filaments (IFs).

IF Type	Method	Failure Force (nN)	Mode	Extensibility	Energy Dissipated/Toughness (MJ m^−3^)
Desmin [7]	AFM	30–40 applied force ^1^	Stretching the filaments adsorbed on a surface	240	-
Keratin (K5/K14) [7]				240	
Neurofilaments [7]				260	
Desmin [8]		3.5		240	50
Vimentin [4]	LOT	0.6 (no failure) ^2^	Axial (end-to-end) stretching		-
Vimentin [9]	AFM	8 (no failure) ^3^	Axial (end-to-end) stretching	300	-
Lamin [10]	AFM	3 (apparent failure)	Pushing perpendicular to the filament axis	250	147
Hagfish slime [11]	Glass microbeam force transducer apparatus	-	Stretching	220	130

AFM = Atomic force microscopy. LOT = Laser optical trap. ^1^ Maximum force applied to observe filament stretching and failure. Exact force where failure occurred was not reported. ^2^ Force applied with LOT was in the sub-nN range. ^3^ AFM is capable of measuring forces reaching tens of nN.

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
