# Peer review of "Bend, Push, Stretch: Remarkable Structure and Mechanics of Single Intermediate Filaments and Meshworks"

_cells, 2021, doi:10.3390/cells10081960_

Round 1
Reviewer 1 Report
This review article by Tanuj Sapra and Medalia is a very interesting and much needed summary and critical analysis of the research field, and proposes important future avenues of research. Because the content is of a high quality, my suggested changes below focus on the presentation.
General comments:
- The style of the language is not Scientific English. For example, page 2; "....may require repair mechanisms such as subunit exchange or reassembly processes to kick in..." , page 6; "....stretching of IFs was kick-started by...., page 12; "...., the experiments do not paint an accurate picture in many cases (for example, the existence of..."
- Many sentences are very long, and often use a passive grammatical form.
- Brackets are extensively used, and also misused when referring to acronyme (the bracket should show the acronym, instead of the full word), and brackets should be avoided as much as possible.
- The language is not specific enough. For example, it is not clear what the concepts "...genetic functions..." in the first sentence of the abstract and the "...all types of IF proteins..."in the 4th sentence of the abstract refers to. Also, the first sentence of the main text; "...crowded cellular environment..." maybe means "intracellular environment..."?, page 9; "...technical constraints", page 11; "...in nature". The lack of specificity is a general problem with the whole text.
- Some figure legends describe the layout etc of the figures, while others describes the content and results. Normally, figure legends should only help the reader to understand the figure, but I understand if the authors, although it is not the standard purpose of figure legends, wish to summarise the content of the literature in the images. However, I would suggest to edit the figure legends so the readers will understand what they look at. For example. For example, in the figure legend of Figure 1 B, it is unclear which change that is referred to. In Figure 2 D, it is no clear what the symbols I, II, II refer to. This is a general problem with the figure legends.
Minor:
- Text within brackets that refers to figures are normally described as (Fig.1) etc and the whole word(figure 1) not written out.
- Some but not all references are in blue colour in the document, so I would suggest the authors to check and ensure that all references are correct and referred to in a consistent manner.
- Figure 2; the text is too small to read.
The text would therefore need extensive language editing by a professional Scientific Language editor prior to acceptance.
Author Response
We thank the reviewer for the insightful comments that were fully addressed (see attached document)

Reviewer 2 Report
The review by Sapra and Medalia gives an excellent, as the title says, insight into the structure and mechanics of IFs. It is well written and organized, discussing various approaches and techniques to study the structural properties of IFs with a main focus on lamins and vimentin. Therefore I can only highly recommend publication in Cells. I found only a few spelling/editing mistakes and one part of a figure could be explained better (see below). All these changes can be made in my opinion during proof reading.
Introduction, first paragraph, last but one sentence: I think it should say "assemble" instead of "assembled" (2x).
Chapter 2, first paragraph, first sentence: "Caenorhabditis" instead of "Ceanorhabditis"
fourth paragraph, fifth sentence: "However, the relevance of such assemblies in is unlikely besides..." I think there is something missing after "in".
Figure 1B: Please explain a bit better which shown structure reflects wt and mutant, respectively.
Chapter 3, heading: "Mechanical Probing of IF Networks in Cells, Nuclei" do you mean in "in Cell Nuclei" or "in Cells and Nuclei"?
Chapter 3: The sentence starting with "The E145K..." just ends with besides. Is there something missing?
Chapter 4, first paragraph: "...are stretched resulting in non-linear strain..." stiffening instead of "...are stretched resulting non-linear strain stiffening..."
second paragraph: "...with high tensile strength of 180 MPa and toughness of 130 MJ/m3." instead of "...with high tensile strength of 180 MPa and toughness 130 MJ/m3."
Chapter 5, second paragraph: "...from 0.17 nm for native vimentin to 1.2 nm for glutaraldehyde-treated vimentin..." instead of "...from 0.17 nm for native vimentin to 1.2 nm to glutaraldehyde-treated vimentin..."
Author Response

(The authors gave the same response as above.)

Reviewer 3 Report
This is an outstanding review covering various aspects of mechanical properties of intermediate filaments. It provides historical perspectives on how mechanics of intermediate filaments came to be known and also details current technologies and future directions in addressing major questions in the field. It is highly recommended that the manuscript be published with the following minor edits:
- Figure 1B should indicate what blue and yellow filaments are.
- Abbreviations such as AFM, MD and LOT should be defined.
- Funding and conflicts of interest should be customized.
Author Response

(The authors gave the same response as above.)

Reviewer 4 Report
In their manuscript “Bend, push, stretch: Insight into the structure and mechanics of single intermediate filaments and meshworks” Sapra and Medalia reviewed structural and mechanical aspects of lamin (mostly) and vimentin (less frequented) intermediate filaments (IFs) with clear focus on in vitro studies. The manuscript addresses a very interesting and topical field. Authors comment wide spectra of approaches and techniques that help to understand the structure and mechanical properties of intermediate filaments, from microscopy-based techniques to single molecule mechanical characterization and elasticity measurement of artificial filaments in vitro. The manuscript is well structured and written, authors covered main papers reflecting the current state of IF research providing thus an informative overview. Although focus of review is clearly on in vitro data, authors could consider to significantly strengthen part covering in vivo experiments (especially chapter 4). Although the review gives overall good impression, I have some additional points concerning the current version, which may require the authors’ further consideration:
1) 2nd paragraph of introduction. Authors review elasticity and stiffness of cytoskeletal fibers and filaments, they compare actin and microtubules with IF. I miss recent publications, mostly on vimentin (e.g. V. Levi 2020 , Guo, Goldman 2019 PNAS etc. – these are not mentioned at all in the paper). Only recent review by Broussard – K. Green 2020 is mentiondd.
2) Chapter 4 “In vitro single molecule mechanical characterization of IFs” represents the weakest part of the review (namely 2nd part appears to be somewhat superficially written). List and description of several AFM-based experiments that show mechanical properties of cytoskeletal IFs – authors mentioned irreversible a-helix to b-sheet transition but they do not comment biological significance or any relevant observation in cells. Should be done even though it’s a bit discussed in the following chapter 5.
3) It is pity that authors resigned on investing creative potential into original figures. Two figures (Fig 1A and 3B, D) were frequented with minor modifications in several other publications (Turgay 2017 Nature; Tanuj Sapra 2020 Nat Com.; de Leeuw 2018 Trends in Cell Biology). However, it is properly indicated that they were adopted from other source.
4) It would significantly improve accessibility of the text if list of abbreviations is included. Abbreviations as MD or FE (force-extension) are not explained in the text and could be incomprehensible for readers outside of the field.
Minor points:
Text should be proofread for typos. E.g. chapter 2, line 1: mechanically vs mechanically or chapter 2, paragraph 4, line 10: “the relevance of such assemblies in vivo is unlikely” vs “the relevance of such assemblies in is unlikely” etc.
Author Response
We thank the reviewer for the comments that were fully addressed (see attached document)

Reviewer 5 Report
The review by Sapria nd Medalia entitled: “Bend, Push, Stretch: Insight into the Structure and Mechanics of Single Intermediate Filaments and Meshworks, is a timely and high quality review”.
The review is well written, well structured and accurately referenced.
There are only few items that require further attention:
Abstract: The sentence “The basic building block of all IFs is an elongated -helical coiled-coil that assembles into complex hierarchical structures and later meshworks.” requires rephrasing.
Introduction: “
In the crowded cellular environment, information passes across distances up to tens of micrometers.” The cell type needs to be specified because the sentence does not apply for skeletal muscle, motor neurones, sensory neurones etc.
“Thus, the ancestors of the IF family of proteins assembled into architecturally simpler filaments.” This is an unreferenced statement that requires in addition a short layout of the supporting data.
“Because bending stiffness and persistence length (lp) are directly related, out of the three cellular filaments” The term persistence length requires the necessary introduction.
“Over 500 reported mutations in the lamin genes, recently found in LMNB [32]” The involved LMNB genes need to be specified.
“Nevertheless, paracrystals do offer an opportunity to study mu-tations; for example, Hutchinson-Gilford Progeria syndrome causing mutation Q159K led to a different assembly of only 2 protofilaments instead of 3 or 4 as in the wild-type lamin [66]. “ Which lamin gene (species) does Q159K belong to? The reviewer is not aware of a Q159K mutation that cause HGPS is humans. Therefore this sentence requires attention by specifying the species involved and by correcting the terms for the disease.
“That allowed to decipher the structural framework of mammalian lamin meshwork and meas-ure the basic physical dimensions at a single lamin filament level. Surprisingly, the ances-tor of all ~10 nm thick IFs assemble into ~3.5 nm thick filaments.” Specify the type of lamin meshwork and provide evidence/references as to why lamins are considered ancestral Ifs.
Figure 1. Panel B, the figure requires further labels. Which segment is WT and which mutant? Is the filament tetramer a WT or mutant protein?
“localization of telomeres and perturbs the lamin network as indicated by lobulated nuclei besides [67].” The sentence requires attention.
“A more striking difference was observed in the Young’s moduli of lamin A (~40–80 MPa) and E145K-lamin A (~100–340 MPa).” Citations are required
“The Young’s modulus of isolated nucleus (~9 kPa) was three-fold lower than that in the cell (~27 kPa) implying that the nucleus is exposed to intracellular balancing forces in vivo, a corner stone of the tensegrity model [83].” The content of the sentence is not very clear.
“Keratinocyte is an important model system for studying the role of keratins in epithelial tissues.” Correct to: The keratinocyte is an important cellular model system for studying the role of keratins in epithelial tissues.
“Developing new methods may also find applications in medical diagnostics;” The types of methods implied require specification.
“Interestingly, keratin cytoskeleton is deformed less when keratinocytes are on stiff sub-strates (70 and 214 kPa) compared to a soft one (8 kPa), and the Young’s modulus (stiff-ness) shows a progressive increase with substrate stiffness [89].” Specification of the keratin cytoskeleton is required.
“The vimentin filamentous network forms a protective cage around the nucleus conferring mechanical stability during migration through tiny pores [84].” Elaborate on tiny pores; the content is not very clear.
“In agreement, MD simulations” MD stands for?
Figure 2 legend; Requires expansion in order to aid understanding, as an example, panel A cannot be understood with the details given.
Rephrase the title of Figure 3. “Pushing lamin filaments in native meshwork.”
The content of the sentence “Lamin filaments deformed to ~90 nm before failing in steps of 4 nm and 8 nm at forces ranging from 2 nN to 5 nN. In agreement, force-clamp experiments showed that single lamin filaments in a network mainly failed in discrete steps of 1, 4 and 8 nm, and can withstand forces of at least 3 nN [106].” Is not very clear.
“Keratin in keratinocytes, lamin concentrations are regulated depending on the develop” Keratins in keratinocytes…
“Lamin A is pre-dominantly present in ‘harder’ fibroblasts and muscle cells but absent from red blood cells, and its thickness layer is controlled in different cell types and physiological state of the cell [148].” Define the nature of red blood cells. In humans, red blood cells do not contain a nucleus.
The review will benefit from a table that summarizes the key similarities/differences (values; AFM parameters measured, etc) with the respect to the biomechanical properties of IFs.
Author Response
We thank the reviewer for the comments which were fully addresed (see attached file)
